# Whey Protein Isolate Nanofibers Prepared by Subcritical Water Stabilized High Internal Phase Pickering Emulsion to Deliver Curcumin

**DOI:** 10.3390/foods11111625

**Published:** 2022-05-31

**Authors:** Xin Xu, Zhiyi Zhang, Junlong Zhu, Dan Wang, Guoyan Liu, Li Liang, Jixian Zhang, Xiaofang Liu, Youdong Li, Jiaoyan Ren, Qianchun Deng, Chaoting Wen

**Affiliations:** 1College of Food Science and Engineering, Yangzhou University, Yangzhou 225127, China; xuxin@yzu.edu.cn (X.X.); zhzhyi0928@163.com (Z.Z.); z17502110240@163.com (J.Z.); wangdan683280@163.com (D.W.); yzufsff@163.com (G.L.); liangli0508@hotmail.com (L.L.); zjx@yzu.edu.cn (J.Z.); liuxf@yzu.edu.cn (X.L.); 008040@yzu.edu.cn (Y.L.); 2School of Food Science and Engineering, South China University of Technology, Guangzhou 510641, China; jyren@scut.edu.cn; 3Oil Crops Research Institute (OCRI), Chinese Academy of Agricultural Sciences, Wuhan 430062, China; dengqianchun@caas.cn

**Keywords:** subcritical water, whey protein isolate nanofibers, high internal phase Pickering emulsion, curcumin deliver system, bioavailability

## Abstract

This study aimed to design a Pickering emulsion (PE) stabilized by whey protein isolate nanofibers (WPINs) prepared with subcritical water (SW) to encapsulate and prevent curcumin (Cur) degradation. Cur-loaded WPINs–SW stabilized PE (WPINs–SW–PE) and hydrothermally prepared WPINs stabilized PE (WPINs–H–PE) were characterized using the particle size, zeta potential, Congo Red, CD, and TEM. The results indicated that WPINs–SW–PE and WPINs–H–PE showed regular spherical shapes with average lengths of 26.88 ± 1.11 μm and 175.99 ± 2.31 μm, and zeta potential values were −38.00 ± 1.00 mV and −34.60 ± 2.03 mV, respectively. The encapsulation efficiencies of WPINs–SW–PE and WPINs–H–PE for Cur were 96.72 ± 1.05% and 94.07 ± 2.35%. The bio-accessibility of Cur of WPINs–SW–PE and WPINs–H–PE were 57.52 ± 1.24% and 21.94 ± 2.09%. In addition, WPINs–SW–PE had a better loading effect and antioxidant activities compared with WPINs–H–PE. SW could be a potential processing method to prepare a PE, laying the foundation for the subsequent production of functional foods.

## 1. Introduction

Curcumin (Cur) is found in the rhizome of turmeric (*Curcuma longa*) and is a characteristic polyphenolic compound. Cur has antioxidant, anti-inflammatory, and anticancer properties, among others [1,2,3]. The U.S. Food and Drug Administration (FDA) generally considers Cur safe at low levels [4]. Cur is a hydrophobic molecule and practically insoluble in water, inhibiting its application in medicines and health products [5]. Therefore, a delivery system is needed to promote the water dispersibility, chemical stability, and bioavailability of Cur [6,7]. The nano-system enhances the biological effects of drug ingestion by protecting the drug from enzymatic degradation, providing a controlled release and altering residence time, among others [8]. Delivery systems for Cur nano-formulations include emulsion, liposomes, solid lipid nanoparticles, polymer nanoparticles, polymer micelles, etc. Emulsions are divided into two main categories, including traditional emulsion and Pickering emulsion (PE). Recently, PE has attracted much attention in delivery systems due to its use of solid particles to stabilize oil droplets, with higher thickness and surface loading than traditional emulsion [9]. PE can protect hydrophobic bioactive substances and deliver them to target sites due to their good chemical and physical stability [10]. In addition, PE includes traditional PE and high internal phase PE (HIPPE). The minimum internal ratio of HIPPE was 0.74. HIPPE can be prepared with only a few stabilizers, which have strong anti-coalescence ability and storage ability. HIPPE has significant advantages as a delivery system, with strong stability (chemical, environmental, gastrointestinal stability, etc.) and high bioavailability for delivering bioactive substances. Since the interface is occupied by most of the particles, the interfacial tension is reduced [11]. It is worth noting that food-grade macromolecules (polysaccharide, protein, polyphenol, etc.) were used to stabilize HIPPE with significant advantages, especially for animal-derived protein [12,13,14,15]. Therefore, the selection and development of suitable animal protein stabilizers to prepare HIPPE are of great significance for improving the utilization rate of biologically active substances.

Whey protein isolate (WPI) is a functional food with high nutritional value and contains more essential amino acids than plant protein [16]. Some studies have found that the heated protein is a good stabilizer for emulsion [17]. Thermal treatment techniques can alter the emulsification and functional properties of protein [18]. The tertiary structure of the protein is partially or fully unfolded upon heating, resulting in the exposure of hydrophobic groups, thereby increasing its flexibility [19]. Whey protein isolate nanofibers (WPINs) are nanofibers formed by heat-induced denaturation of protein [20]. Fibrosis usually begins with globular protein complete or partial unfolding and ends with an ordered fibrous structure [21]. Studies have shown that WPINs can be prepared via heating in a water bath at 80 °C for 10 h [22]. WPINs can also be prepared by using an oven reaction kettle at 110 °C for 4 h [23]. These two traditional methods (heat treatment method, oven reaction kettle treatment method) for preparing WPINs were time-consuming, energy-intensive, and difficult for industrial production. Therefore, researchers began to adopt new technologies, such as the subcritical water (SW) treatment method. In the SW treatment method, the liquid is heated to 100–374 °C under a certain pressure [24]. This method is very environmentally friendly and efficient. A previous study confirmed that the *Lycium barbarum* polysaccharide-protein conjugates obtained under SW conditions (120 °C) can be used for stabilized selenium nanoparticles and exhibit good activity [25]. Another study also explored the synthesis of nano-catalysts with SW at 100–500 °C and found that the optimal synthesis temperature of different chemical materials was different [26]. Compared with natural heat-treated soybean protein (90 °C), SW (>100 °C) extraction of protein from heat-denatured soybean exhibits excellent interfacial properties and higher surface activity [27]. Studies have shown that the high temperature and pressure in SW conditions are more conducive to the degradation of cellulose and the preparation of WPINs, resulting in a shorter preparation time [28]. As far as we know, the preparation of WPINs stabilized HIPPE from SW treatment has not been performed.

This study aimed to efficiently prepare WPINs in a short time using SW (pressure: 0.2 MPa). WPINs (c = 5 wt%) and corn oil (Φ = 0.74) were used to prepare the HIPPE. Cur was embedded in HIPPE to explore its bioavailability and antioxidant activity after simulated digestion in vitro.

## 2. Materials and Methods

### 2.1. Materials and Chemicals

Whey protein isolate (>80%) and curcumin (>95%) were provided by Yuanye Biotechnology Co., Ltd. (Shanghai, China). Corn oil was purchased from Suguo supermarket (Yangzhou, China) without further purification. Congo Red, Sodium dodecyl sulfate (SDS), 1,1-diphenyl-2-picryl-hydroxyl (DPPH), 2,2-Azino-bis (3-ethylbenzothiazoline-6-sulphonic acid) diammonium salt (ABTS), and 2,4,6-tripyridyl-s-triazine were purchased from Shanghai Aladdin Biochemical Technology Co., Ltd. (Shanghai, China). FITC, Nile red, and porcine pancreatic lipase (100–650 μ/mg) were obtained from Sigma-Aldrich (St. Louis, MO, USA). Bile salts and porcine pepsin (12 μ/mg) were purchased from Sunson (Beijing, China). All other reagents were analytical grade. All solutions were prepared using purified deionized (DI) water.

### 2.2. Preparation of Whey Protein Isolate Nanofibers by Subcritical Water

The pretreatment steps of the 5% WPI solution (*w*/*w*) refer to the experiment of Yang et al. [23]. In brief, the pH of the WPI solution (5%, *w*/*w*) was adjusted to 2.0 with 3 M hydrochloric acid, stirred at 25 ± 1 °C for 30 min, then centrifuged at 4 °C and 10,000× *g* for 15 min (L550 Cence, Changsha, China). Undissolved protein was removed by vacuum filtration of the supernatant with a filter (0.22 μm pore size, Xinya, Shanghai, China). The filtrate was sent to an SW reactor, and the oven temperature was set to 110 °C. After heating for 5 min and setting 0.2 MPa, 0.5 MPa, 1.0 MPa, and 1.5 MPa pressure, the SW reaction kettle was cooled with an ice bat to obtain WPI nanofibers (WPINs–SW) and stored in a refrigerator at 4 °C. The WPI nanofibers (WPINs−H) prepared by the hydrothermal method were incubated in a water bath at 80 °C for 10 h as a control.

### 2.3. Structural Characterization of WPINs

#### 2.3.1. Particle Size and Zeta Potential

The particle and zeta potential measurements of samples were based on the previous method with some modifications [7]. The particle size of WPINs was measured using a Malvern Nano sizer (Mastersizer 1000, Malvern Instruments Ltd., Malvern, UK). Samples were diluted 1000 times by the distilled water and 3200 rpm vortex vibrations for 1 min before measurement.

Zeta potential Analyzer (Mastersizer 1000, Malvern Instruments Ltd., Malvern, UK) was used to measure the zeta potential of the nanoparticles. The samples were diluted 100 times by the distilled water and were added to a cuvette equipped with an electrode.

#### 2.3.2. Congo Red Binding Spectrum

The Congo Red binding capacity measurement of samples was in accordance with the method of Nilsson et al. [29]. After mixing 500 μL of the sample solution with 5 mL of the Congo Red solution (70 μg/mL, pH 7.0, 10 mM phosphate buffer), we let it stand for 15 min at room temperature. The spectra were then observed at 400–600 nm with a photometer (Cary 5000, Varian, Palo Alto, CA, USA).

#### 2.3.3. Circular Dichroism (CD) Spectroscopy

The secondary structural changes of WPINs were measured according to the previously described methods using CD spectroscopy [30]. CD spectra were observed with a Jasco spectropolarimeter (Model J-810, Jasco, Tokyo, Japan) in the far ultraviolet (190–260 nm) region and at room temperature. The final protein concentration of the WPINs solution was diluted to 0.25 mg/mL. Each scan was repeated three times and averaged.

#### 2.3.4. Transmission Electron Microscopy (TEM)

According to the method of Wang et al. [22], the concentration was diluted to 0.1%. Then, the WPINs solution was slowly dropped onto a specially made copper mesh (with a diameter of 3 mm and a thickness of 10–30 µm). After standing for 15 min, we gently blotted off excess liquid with filter paper. Then, 2% uranyl acetate was added dropwise to a dry mesh, and the copper was left to ventilate for 8 min. Before TEM measurement, the mesh was absorbed again with filter paper to remove the unwanted solution. Electron micrographs were taken by an HT7800 transmission electron microscope (Hitachi, Tokyo, Japan).

### 2.4. Preparation and Characterization of High Internal Phase Pickering Emulsion

HIPPE was prepared by homogenizing WPINs solution (5%, *w*/*w*) with a fixed oil–water ratio of 7.4:2.6 (mL/mL). The suspension (2.6 mL) with different WPINs concentrations was mixed with corn oil (7.4 mL) using a shear emulsifying machine (T18, IKA, Staufen im Breisgau, Germany) at 20,000 rpm for 5 min. WPINs–SW–PE was prepared with WPINs (SW treatment)-stabilized PE, while WPINs–H–PE was prepared with WPINs (hydrothermal treatment)-stabilized PE.

### 2.5. Structural Properties of Pickering Emulsion

#### 2.5.1. Optical Microscopy

Photographs refer to the method of Ren et al. with modifications [31]. The morphology of the PE was observed by an optical microscope (Motic, Panthera LBA310-T, Hong Kong, China). We diluted HIPPE by a factor of 5 and obtained photographs using 100×, 400×, and 1000× lenses.

#### 2.5.2. Confocal Laser Scanning Microscopy (CLSM)

The adsorption of nanoparticles at the oil–water interface can be observed by confocal laser scanning microscopy (CLSM) (LSM 880 NLO, Carl Zeiss AG, Ober-kochen, Germany). PE was diluted by 100 times before shooting. The emulsion was further characterized with the method of Li et al. with minor modifications [32]. The oil phase was stained with Nile red, and the protein phase was stained with FITC during the emulsion preparation. Emulsions were observed under a 10× objective. The laser excitation source for Nile red was 488 nm and the laser excitation source for FITC was 543 nm.

#### 2.5.3. Particle Size and Zeta Potential

The PE size was measured by the Malvern Nano sizer (Mastersizer 3000, Malvern Instruments Ltd., Malvern, UK). The Zeta Potential Analyzer (Mastersizer 1000, Malvern Instruments Ltd., Malvern, UK) was used to measure the zeta potential of the nanoparticles. The samples were diluted 1000 times by the distilled water and were added to a cuvette equipped with an electrode.

#### 2.5.4. Emulsifying Capacity (EAI) and Emulsifying Stability (ESI)

According to the previous method of de Souza et al., the evaluation of emulsifying performance mainly relied on emulsifying ability (EAI) and emulsifying stability (ESI) [33]. The PE preparation was performed using the same procedure as Section 2.4. The freshly prepared emulsion (50 μL) was diluted with 0.1% SDS, and the absorbance at 500 nm was measured (recorded as A_0_). After standing at room temperature for 10 min, the absorbance of the diluted emulsion was measured again (denoted as A_10_). EAI and ESI were calculated according to Equations (1) and (2).
(1)$$\mathrm{EAI}\;(m^{2}/g)=\frac{2\;\times \;2.303\;\times \;A_{0}\;\times \;\mathrm{DF}}{C\;\times \;\emptyset \;\times \;\theta \;\times \;10000}$$
(2)$$\mathrm{ESI}\;(\min)=\frac{\;A_{0}\;\times \;10}{\Delta A}$$

DF, C, and ∅ represent the dilution factor, protein concentration, and oil volume fraction, respectively. θ represents the optical path (0.01 m). ΔA represents the difference between A_0_ and A_10_.

#### 2.5.5. Centrifugation Stability

The measurement of centrifugal stability employed the method of Gond et al. [34]. After preparing the PE, an appropriate amount of the emulsion was immediately taken out and placed in a 10 mL centrifuge tube. After centrifugation at a centrifugal force of 10,000× *g* for 10 min, the emulsion oil–water separation and particle precipitation were compared. 

### 2.6. Preparation of Curcumin-Loaded Pickering Emulsion

Cur was added to corn oil at a concentration of 0.1 wt%. To ensure the maximum solubility of Cur in corn oil, the mixture was stirred overnight under magnetic stirring (800 rpm). The undissolved Cur in the mixture was then removed by centrifugation at 10,000× *g* for 10 min. The supernatant was used as an oil phase in the SW prepared WPINs to stabilize PE (WPINs–SW–PE–Cur) and hydrothermal-method-prepared WPINs to stabilize PE (WPINs–H–PE–Cur), which was prepared using the same procedure as Section 2.4.

#### 2.6.1. Embedding Rate of Curcumin

The embedding rate of Cur was measured according to the method of Han et al. [35] Freshly prepared Cur-loaded PE was centrifuged at 10,000× *g* for 10 min to remove any large particles and nonencapsulated Cur crystals and dissolved in ethanol. The embedding rate of the PE was then calculated using Equation (3):(3)$$\mathrm{Embedding}\;\mathrm{rate}=\frac{\mathrm{mass}\;\mathrm{of}\;\mathrm{Cur}\;\mathrm{in}\;\mathrm{PE}}{\mathrm{total}\;\mathrm{mass}\;\mathrm{of}\;\mathrm{Cur}}\;\times \;100\%,$$

The concentration of PE can be tested by measuring the absorbance at 425 nm with a UV spectrophotometer (TianMei UV100, Shanghai, China). We drew a standard curve by measuring the absorbance of a series of known concentrations of Cur in ethanol. We diluted the oil phase in ethanol and measured the absorbance at 425 nm recovery of Cur from PE by demulsification with ethanol. Briefly, 100 μL of the emulsion was added to 900 μL of ethanol, and the mixture was centrifuged at 10,000× *g* for 1 min to pellet WPINs. After centrifugation, the supernatant was diluted 10 times with ethanol. We then converted the absorbance of the diluted ethanolic extract to the Cur concentration according to the standard curve.

### 2.7. Release of Curcumin from the Pickering Emulsion during Digestion In Vitro

Before performing in vitro digestion experiments, two mock digestion solutions were prepared. Artificially simulated gastric fluid (SGF; 34.22 mM NaCl, 226.11 mM hydrochloric acid) and simulated intestinal fluid (SIF; 3.75 M NaCl, 249.49 mM calcium chloride dihydrate) were prepared. Bile salts are difficult to dissolve completely, so they were dissolved in a buffer solution (pH 7.0, 53.57 g/L) 24 h before digestion in the small intestine. All mock digests were preheated to 37 °C before each digestion stage.

The lipid content of all samples was adjusted to 2%. Pepsin (0.064 g) was dissolved in 20 g of SGF to obtain the gastric phase electrolyte. Furthermore, 20 g of the emulsion was mixed with 20 g of SGF. The solution was preheated to 37 °C. We adjusted the pH of the mixture to 2.5 with HCl and spun it at 37 °C for 2 h at a stirring speed of 100 rpm.

Chyme samples (30 g) obtained from the final stage were placed in a constant temperature water bath at 37 °C and the pH was adjusted to 7.0. While stirring, we added 1.5 mL of SIF and 3.5 mL of the bile salt solution, prepared in advance, to the chyme sample while adjusting the pH of the mixture to 7.0 for a second time. We then added 2.5 mL of porcine pancreatic lipase under continuous stirring (24 mg/mL), the temperature was maintained at 37 °C, and it was rotated at a stirring speed of 100 rpm for 2 h [36].

#### 2.7.1. Bio-Accessibility

The fraction dissolved in gastrointestinal fluids can generally be seen as the bioavailability of Cur in vitro [37]. Therefore, the amount of released Cur from various formulations was measured in the gastric and small intestine stages. Cur released from the gastric phase was measured after the samples were incubated in simulated gastric fluid for 2 h. In contrast, in the small intestine phase, it was measured after 2 h of incubation of the models in simulated small intestinal fluid. Finally, the samples were centrifuged at 5000× *g* for 30 min, and the supernatant was collected and diluted in ethanol to determine the release of Cur (Section 2.6.1). The bio-accessibility of Cur was calculated in vitro using Equation (4):(4)$$\mathrm{Bio}-\mathrm{accessibility}=\;\;\frac{\mathrm{Weight}\;\mathrm{of}\;\mathrm{solubilized}\;\mathrm{Cur}}{\mathrm{Weight}\;\mathrm{of}\;\mathrm{Cur}\;\mathrm{before}\;\mathrm{digestion}}\times 100\%,$$

It should be noted that this expression for bio-accessibility depends on the fraction of Cur lost due to chemical degradation and the solubilized fraction within the micelle phase.

#### 2.7.2. Curcumin Release

Samples were taken at 120 min in the simulated stomach and 240 min in the intestine, diluted with ethanol, and measurements were performed with a UV spectrophotometer with a wavelength of 425 nm.

### 2.8. Determination of the Antioxidant Capacity

The antioxidant capacity of different Cur formulations (encapsulated and free) was measured following simulated gastrointestinal digestion in vitro.

#### 2.8.1. DPPH• Scavenging Capacity

This spectrophotometric assay uses the stable free radical 1,1-diphenyl-2-picryl-hydroxyl (DPPH) as a reagent. To evaluate the DPPH• scavenging ability of Cur, the method of Gulcin was used with slight modifications [38]. The DPPH• absorbs at 517 nm, but its absorption decreases after reduction by antioxidants or radical species. Based on this principle, we prepared a 0.1 mM DPPH solution in ethanol and added 0.5 mL to 1.5 mL of Cur ethanol solutions of various concentrations (15–45 ug/mL). The mixed solution was vortexed for 3 min and incubated in the absence of light for 30 min. The antioxidant capacity of the control group was determined by a blank sample containing no scavenger and then subtracted from the antioxidant capacity of the corresponding model containing Cur. Controls included free Cur (Free–Cur) simulated gastrointestinal fluid, Free–Cur biopolymer-coated WPINs (Cur for encapsulation) exposed to simulated gastrointestinal fluid, and a Free–Cur physical mixture exposed to simulated gastrointestinal fluid (for non-encapsulated Cur).

#### 2.8.2. ABTS·+ Scavenging Capacity

The ABTS·+ scavenging assay is based on a previously described method [39]. ABTS was dissolved in deionized water to a concentration of 7 mM one day in advance, then 2.45 mM potassium persulfate was dissolved in the ABTS solution and stored at room temperature in the dark. The ABTS solution was then diluted with 10 mM phosphate buffer (pH 7.4) to an absorbance of 0.70 ± 0.01 cm^−1^ at 734 nm and equilibrated at 30 °C. The Cur preparation was centrifuged through simulated GIT at 5000× *g* for 30 min and the supernatant was collected. We adjusted the Cur concentration of the supernatant with phosphate buffer to achieve 20–80% inhibition of blank absorbance by pre-assay. We then dissolved 4.0 mL of sample in 40 mL of the ABTS solution and took absorbance readings at 30 °C after 7 min of initial mixing. Refer to 2.8.1 for the rest of the steps.

#### 2.8.3. Determination of Reducing Power

The FRAP assay was performed with some modifications [40]. The FRAP reagent was prepared with FeCl_3_·6H_2_O (20 mmol/L), 2,4,6-tripyridyl-s-triazine (10 mmol/L), and acetate buffer (300 mmol/L, pH 3.6.) at a ratio of 1:1:10.

The stand curve was drawn using various concentrations of FeSO_4_·7H_2_O (100–800 µM). Cur solution (100 µL) was added to 3 mL of the FRAP reagent. The mixed reagents were placed in a constant temperature water bath at 37 °C for 30 min. The absorbance was measured at a wavelength of 539 nm using a UV spectrophotometer (TianMei UV100, China). FRAP values are expressed as the ratio of Fe^2+^ per μmol to sample per mg (μmol/mg).

### 2.9. Statistical Analysis

All experiments were performed in triplicate, and the results were reported as the mean ± standard. Statistical analysis was performed using the software IBM SPSS Statistics 26.0. Data were subjected to analysis of variance, and the Turkey HSD test was used to determine the significance of differences between data values (*p* < 0.05).

## 3. Results and Discussion

### 3.1. Effects of Subcritical Water on the Structural Properties of WPINs

#### 3.1.1. Particle Size and Zeta Potential of WPINs

Numerous studies report that protein nanoparticles were widely used to stabilize PE due to their high stability. The particle size and zeta potential of the nanoparticles were the main indicators for assessing the structural properties of the nanoparticles. To study the effects of SW pressure on the fibrosis degree of WPI, the particle size and zeta potential of WPINs were characterized. As shown in Figure 1A, different SW pressures can affect the particle size of WPINs. The order of particle size from largest to smallest was WPINs–SW (1.5 MPa), WPINs–SW (0.5 MPa), WPINs–SW (1.0 MPa), and WPINs–SW (0.2 MPa), of which the values were 232.40 nm, 225.44 nm, 222.26 nm, and 190.90 nm, respectively. Compared with other SW pressures, the nanoparticles treated with 0.2 MPa had the smallest particle size, which showed that SW significantly promoted the degree of protein fibrosis, which was conducive to the formation of smaller particles, possibly related to the structural changes of the protein [41]. In addition, the particle size of nanofibers increased with the increase in pressure, which might be due to the excessive pressure promoting the aggregation of protein fibers, and it was not conducive to the preparation of emulsion. Similar results were also reported. For example, the increase in protein particle size was due to the aggregation of nanoparticles caused by excessive pressure [28]. Similarly, the zeta potential of the samples had similar trends, and there were no significant differences between the samples (Figure 1B). Therefore, WPINs–SW (0.2 MPa) was chosen as the optimum SW pressure for subsequent nanofiber preparation.

As shown in Figure 1C, the particle size of WPI, WPINs–SW, and WPINs–H was 243.50 nm, 190.90 nm, and 218.90 nm, respectively. Compared with WPI, the particle sizes of WPINs–SW and WPINs–H decreased by 52.60 nm and 24.60 nm, respectively. The zeta potential of WPI, WPINs–SW, and WPINs–H was 23.50 mV, 26.37 mV, and 27.20 mV, respectively. The zeta potential of WPINs–SW and WPINs–H was considerably higher than that of WPI. The PE prepared by WPINs–SW was more stable, and WPINs–H had more significant zeta potential.

#### 3.1.2. Congo Red Binding Spectrum

Congo Red can specifically bind to nanofibers, and the absorption intensity and absorption peak of Congo Red was increased. The change in absorption intensity can characterize the degree of WPI fibrosis [42]. As shown in Figure 1D, the absorption curve of Congo Red is the black curve, and the absorption curves of Congo Red, which are bound to WPINs–H, WPINs–SW, and WPI, are the red, blue, and green curves. When Congo Red reacted with WPINs, the absorption peak shifted from 490 nm to 540 nm. The order of the absorbance magnitude was WPINs–SW > WPINs–H > WPI, and the absorbance increased with mounting pressure, likely because the pressure caused more nanofibers to be formed and aggregated. This indicated that pressure was a factor affecting the generation of WPINs. In addition, the absorption intensity of WPINs–SW for Congo Red was much higher than that of WPINs–H. It indicated that the appropriate pressure promoted the formation of WPINs. This result was similar to the previous report that the zein structure was altered by subcritical water [43]. The results show that the high temperature and acidic environment caused the protein secondary structure to open, the protein was denatured, and protein particles became nanofibers.

#### 3.1.3. Circular Dichroism (CD) Spectroscopy

Far-ultraviolet CD spectra (190–240 nm) can reflect the secondary structure of WPINs, including α-helix, β-sheet, β-turn, and random coil conformation [44]. For WPINs and WPI, the CD spectra had negative peaks near 190–280 nm and partial α-helix structure peaks had negative peaks at 208 nm and 222 nm. The average residual ellipticity at 216 nm may represent the content of the β-sheet structure. In Figure 1E, the β-sheet structure of WPINs–SW and WPINs–H were significantly more than the β-sheet structure of WPI, indicating that the formation of the β-sheet secondary structure promoted the formation of WPINs. The β-sheet structure of WPINs–SW was more than the β-sheet structure of WPINs–H. Under acidic conditions, some proteins were hydrolyzed into polypeptide fragments due to high-temperature denaturation, and their secondary structures were expanded. Exposure of hydrophobic groups in proteins enhanced intermolecular hydrophobic interactions and assembled into fibers [45]. The formation of the β-sheet during SW was due to the reconstruction of a stable native secondary structure [46]. This result was consistent with the Congo Red analysis.

#### 3.1.4. Transmission Electron Microscopy

WPI can form rough protein nanotube aggregates after a period of thermal induction treatment, and the physicochemical properties of the fibers changed significantly. The length and morphology of nanotube aggregates formed by different preparation methods were significantly different [47]. As shown in Figure 1F, WPINs–SW was a straight linear aggregate. It had many branches and individual protein particles. WPINs−H was sparse with short chapters. Fibrosis occurred in both WPINs–SW and WPINs–H. At pH 2.0, the surface of WPI had a higher charge, and there was still electrostatic repulsion between WPI molecules, resulting in the formation of WPINs [48]. SW denatured the protein, which unfolded the structure and exposed hydrophobic groups, and enhanced hydrophobic interactions between adjacent molecules [45]. It was shown that the average length of protein fiber aggregates increased with increasing pressure. This result proved that pressure is one of the reasons for the WPI fibrosis. Pressure might promote the fibrillation of WPI, causing PE prepared from WPINs to be more stable.

### 3.2. Effects of Subcritical Water on the Properties of High Internal Phase Pickering Emulsion Stabilized by WPINs

#### 3.2.1. Optical Microscopy

In Figure 2A, the WPINs–SW–PE formed spherical aggregates with uniform size distribution. Since the WPINs–SW was closely arranged on the surface of the oil droplets, a large amount of electrostatic repulsion was provided between the PE particles, which made the WPINs–SW–PE very dense and oil droplets disperse independently [49]. However, the particle sizes of WPINs–H–PE were different, whereby one part was dispersed and the other part was aggregated, which meant that the oil droplets were loosely wrapped by WPINs–H, and the oil droplets were aggregated together. The particle size of WPINs–SW–PE was significantly smaller than that of WPINs–H–PE, and the result was consistent with the results of the particle size measurement. The WPINs–SW–PE under 0.2 MPa was more uniform. The emulsion had good stability and a good effect on the subsequent embedding of Cur.

#### 3.2.2. CLSM

The mechanism of PE stability can be observed by CLSM photography. Under the fluorescent field, the protein bound specifically to FITC and emitted green light. The oil phase was combined explicitly with Nile red and glowed red. As shown in Figure 2B, the red oil phase particles were covered by green nanoparticles, indicating that the emulsion type was an oil-in-water (O/W) emulsion. A red ring layer protected the surface of oil particles, and FITC-stained protein was adsorbed at the oil–water interface and provided a barrier for the coalescence of oil particles [50]. This finding provided the most intuitive evidence for the mechanism as WPINs stabilized for PE. The emulsion did not show significant agglomeration. The oil–water interface presented a yellow ring layer caused by the superposition of the green color of FITC and the red color of Nile red during CLSM imaging, indicating the presence of green-stained protein at the oil–water interface. The CLSM images of WPINs–SW–PE showed that WPINs–SW was completely adsorbed around the oil droplets and dispersed in the spaces between the oil droplets. The photos of WPINs–SW–PE showed no aggregation and flocculation of oil droplets, and the number of particles at the interface was sufficient for protein adsorption. There was a slight aggregation of oil droplets in WPINs–H–PE, indicating that WPINs–H did not wrap oil droplets well and WPINs–H did not emulsify as well as WPINs–SW.

#### 3.2.3. Particle Size and Zeta Potential

Particle size and zeta potential were important parameters for judging the stability of particles to the PE. The average particle size and zeta potential of the WPINs stabilized HIPPE are shown in Figure 2C. The particle size of WPINs–H–PE was 175.99 μm. The particle size of WPINs–SW–PE was 26.88 μm. SW led to a reduction in the average particle size. In addition, the zeta potential was used to represent the electrostatic repulsion between charged particles. The absolute value of the high zeta potential indicated the high stability of PE [51]. The absolute value of WPINs–SW–PE on the zeta potential was greater than that of WPINs–H–PE on the zeta potential. Interestingly, both of them were greater than 30 mV, indicating that WPINs treated with both SW and hydrothermal methods could enhance the stability of PE [52]. Based on these results, SW treatment may reduce the average particle size and increase the zeta potential of PE, which was attributed to the increase in the surface charge and distribution of WPINs during SW. WPINs–SW–PE will act as a higher and more stable encapsulation rate during the subsequent encapsulation of Cur.

#### 3.2.4. Emulsification Activity Index (EAI) and Emulsification Stability Index (ESI)

The emulsification activity index (EAI) and emulsification stability index (ESI) of PE were determined. The EAI value was the maximum interfacial area per unit weight of protein in a stable solution. The ESI indicated the stability of a diluted emulsion over a specified period [53]. As shown in Figure 2D, the EAI of WPINs–SW–PE was 186.63 m^2^/g, and the EAI of the WPINs–H–PE was 151.79 m^2^/g. The hydrophobic and electrostatic interactions between droplets were increased because the protein mass ratio was 5%, so the WPINs exhibited the best emulsification activity [49]. The fibrosis of WPI became more complete under the pressure of SW, and the emulsifying activity of PE was increased. Meanwhile, the ESI of the WPINs–H–PE was 89.05 min, and the ESI of WPINs–SW–PE was 670.30 min. PE can maintain a relatively stable state because of the strong electrostatic repulsion between molecules, which was conducive to the adsorption of WPINs on the oil–water interface. Besides, the emulsion showed no demulsification phenomenon, indicating that the particles adsorbed on the interface had higher emulsification stability [54]. Significant emulsification activity and emulsification stability of the PE were a function of the excellence of the water phase particle, which provided a more stable emulsion option for the embedded Cur below.

#### 3.2.5. Centrifugation Stability

As shown in Figure 2E, the water–oil ratio of the HIPPE was 2.6:7.4 after centrifugation at 10,000× *g* for 10 min. The oil phase was not precipitated, and no sediment particles were observed in the lower water phase. It may be that HIPPE was highly cohesive and able to form a dense network structure, while PE was hard and did not flow easily. The particles can be firmly adsorbed by the interface to prevent the emulsion from merging, resulting in ultra-high stability of the emulsion. Most PE prepared with protein-based particles was difficult to maintain under centrifugal forces of more than 5000× *g* [34,55,56].

### 3.3. Effects of High Internal Phase Pickering Emulsion on Curcumin Bioavailability

#### 3.3.1. Embedding Rate of Curcumin

As shown in Figure 3A, the embedding rate of WPINs–SW–PE–Cur was 96.72%, and the embedding rate of WPINs–H–PE–Cur was 94.07%. The results showed that the PE was stabilized by WPINs, and the stabilized particles were adsorbed and arranged at the oil–water interface, which helped improve the embedding rate of Cur. The embedding rate of Cur in WPINs–SW–PE was higher than that of WPINs–H–PE because the embedding of oil droplets by WPINs–SW was stronger than that of WPINs−H, and the structure of WPINs prepared by SW was more stable.

#### 3.3.2. Bio-Accessibility

The release of Cur occurred in the small intestinal stage, which was a prerequisite for exploring bio-accessibility. The bio-accessibility of Cur-loaded HIPPE after in vitro digestion was explored, and the results are presented in Figure 3B. The bio-accessibility of WPINs–SW–PE–Cur was 57.52%, and the bio-accessibility of WPINs–H–PE–Cur was 21.94%. The structure of WPINs–SW was more compact to protect Cur from releasing during simulated gastric digestion in vitro. In addition, the particle size of WPINs–SW–PE was smaller, and the specific surface area in contact with lipase during the simulated small intestinal digestion process in vitro was larger, which caused more Cur to be released and then used by cells, thus causing the higher bio-accessibility of WPINs–SW–PE–Cur [57,58]. However, the particle size of WPINs–H–PE was large, so a large amount of oil cannot be digested by the gastrointestinal tract. Cur was a lipophilic substance, which was dissolved in corn oil, so Cur cannot be released at the target site. The bio-accessibility of WPINs–H–PE–Cur was low.

#### 3.3.3. Curcumin Release

Cur was released in simulated gastric fluid (SGF) and simulated intestinal fluid (SIF). As shown in Figure 3C, after incubation in SGF for 2 h, the Cur release of WPINs–SW–PE–Cur and WPINs–H–PE–Cur reached 25.25% and 29.37%, respectively. After 2 h in SIF, the Cur release of WPINs–SW–PE–Cur and WPINs–H–PE–Cur reached 74.25% and 43.45%, respectively. In the SGF environment, the release of WPINs–H–PE–Cur was higher than that of WPINs–SW–PE–Cur, while the results were reversed in SIF. Perhaps the pH 2.0 gastric acid environment and the presence of pepsin cause the WPINs to be encapsulated in PE hydrolyzed by pepsin, and the electrostatic repulsion between adjacent particles was weakened and the included oil phase was exposed. Since Cur was heavily exposed in the stomach, the amount of Cur reaching the small intestine was reduced [57,59,60]. However, WPINs–SW–PE had good stability, which enabled Cur to be protected in the stomach and achieved a large amount of targeted release in the small intestine. In conclusion, WPINs–SW–PE–Cur significantly improved the bioavailability of Cur. A previous study investigated the release of Cur from stabilized PE encapsulated in a chitosan/gum Arabic antiparticle and noted that approximately 36% of Cur was released after 3 h incubation in SIF [35]. As a comparison, a much higher rate of Cur release can be observed in the current study, which demonstrated that the WPINs–SW stabilized PE had a considerable effect on the targeted release in SIF.

### 3.4. Effects of Simulated Gastrointestinal Digestion on the Antioxidant Capacity of Pickering Emulsion

#### 3.4.1. DPPH• Scavenging Capacity

The DPPH• scavenging capacity of Cur encapsulated in PE was compared to that of Free–Cur after in vitro digestion to understand the effect of protective nano and PE on the antioxidant activity of Cur [61]. As shown in Figure 4A, the DPPH• scavenging capacity of WPINs–SW–PE–Cur, WPINs–H–PE–Cur, and Free–Cur was 48.67%, 38.45%, and 18.43%, respectively. DPPH• scavenging capacity after digestion in vitro was 22.56%, 18.74%, and 7.34%, respectively. It can be seen that the antioxidant activity of encapsulated Cur was significantly higher than that of Free–Cur. The antioxidant activity of WPINs–SW–PE–Cur was significantly better than that of WPINs–H–PE–Cur, which may be related to the solubility of Cur. The solubility of Cur in WPINs–SW–PE was higher, which was more conducive to the full contact between Cur and oxidized substances. The protective effect of PE on the antioxidant activity of Cur was confirmed, thus helping to increase its antioxidant capacity [62].

#### 3.4.2. ABTS·+ Scavenging Capacity

As shown in Figure 4B, the antioxidant activities of WPINs–SW–PE–Cur, WPINs–H–PE–Cur, and Free–Cur were determined. The ABTS·+ scavenging abilities of the WPINs–SW–PE–Cur, WPINs–H–PE–Cur, and Free–Cur were 36.50%, 36.53%, and 12.38%, respectively. After digestion in vitro, the ABTS·+ scavenging abilities were 21.27%, 20.02%, and 5.30%, respectively. The antioxidant activity of Free–Cur was always significantly lower than that of encapsulated Cur, and the ABTS·+ scavenging ability of WPINs–SW–PE–Cur was stronger than that of WPINs–H–PE–Cur. The antioxidant groups of Cur were exposed in the system and combined with ABTS^.^+ in large quantities because PE improved the solubility of Cur and expanded the dispersibility of Cur in the solution. This result was consistent with the DPPH• scavenging ability.

#### 3.4.3. Reducing Power

Figure 4C displays the FRAP iron-reducing ability of WPINs–SW–PE–Cur, WPINs–H–PE–Cur, and Free–Cur before and after simulated digestion in vitro. In the reaction system of the FRAP method, the FRAP iron-reducing ability of the sample was positively correlated with the color and absorbance changes of the final reaction product, which can be used as an essential indicator to measure the antioxidant activity of the sample [63]. Before simulated digestion in vitro, the FRAP iron-reducing ability of Free–Cur was 17.60 μmol/mL, and the FRAP iron-reducing ability of the WPINs–SW–PE–Cur and WPINs–H–PE–Cur was 72.49 μmol/mL and 52.53 μmol/mL. After digestion in vitro, the FRAP iron-reducing ability of the WPINs–SW–PE–Cur and WPINs–H–PE–Cur was 31.52 μmol/mL and 27.91 μmol/mL. The FRAP iron-reducing ability of the Free–Cur was 7.04 μmol/mL, indicating that the PE prepared by WPINs had a good encapsulation effect on Cur and promoted the water solubility of Cur, and the WPINs–SW–PE was more stable. The previous result reported that gelatin-encapsulated Cur showed significantly higher antioxidant activity [64]. In fact, the encapsulation of Cur increased the water solubility of Cur [65].

## 4. Conclusions

This study successfully prepared WPINs–SW at 110 °C and 0.2 MPa in 5 min. TEM images showed that WPINs–SW is composed of fibrous nanoparticles. Short-term high temperature and pressure can allow complete protein modification and acts as a stabilizer for HIPPE. CLSM, optical microscopy, particle size, zeta potential, and EAI and ESI results indicated that the WPINs–SW–PE had a smaller particle size and better stability. The results of simulated intestinal digestion clearly showed that the WPINs–SW–PE had higher Cur release than that of WPINs–H–PE. These findings suggested that WPINs–SW–PE–Cur might be a promising encapsulating agent to protect the loaded Cur and achieve sustainable release under intestinal conditions. However, the absorption and transport functions of active substances delivered by HIPPE remain to be further explored.

## Figures and Tables

**Figure 1 foods-11-01625-f001:**
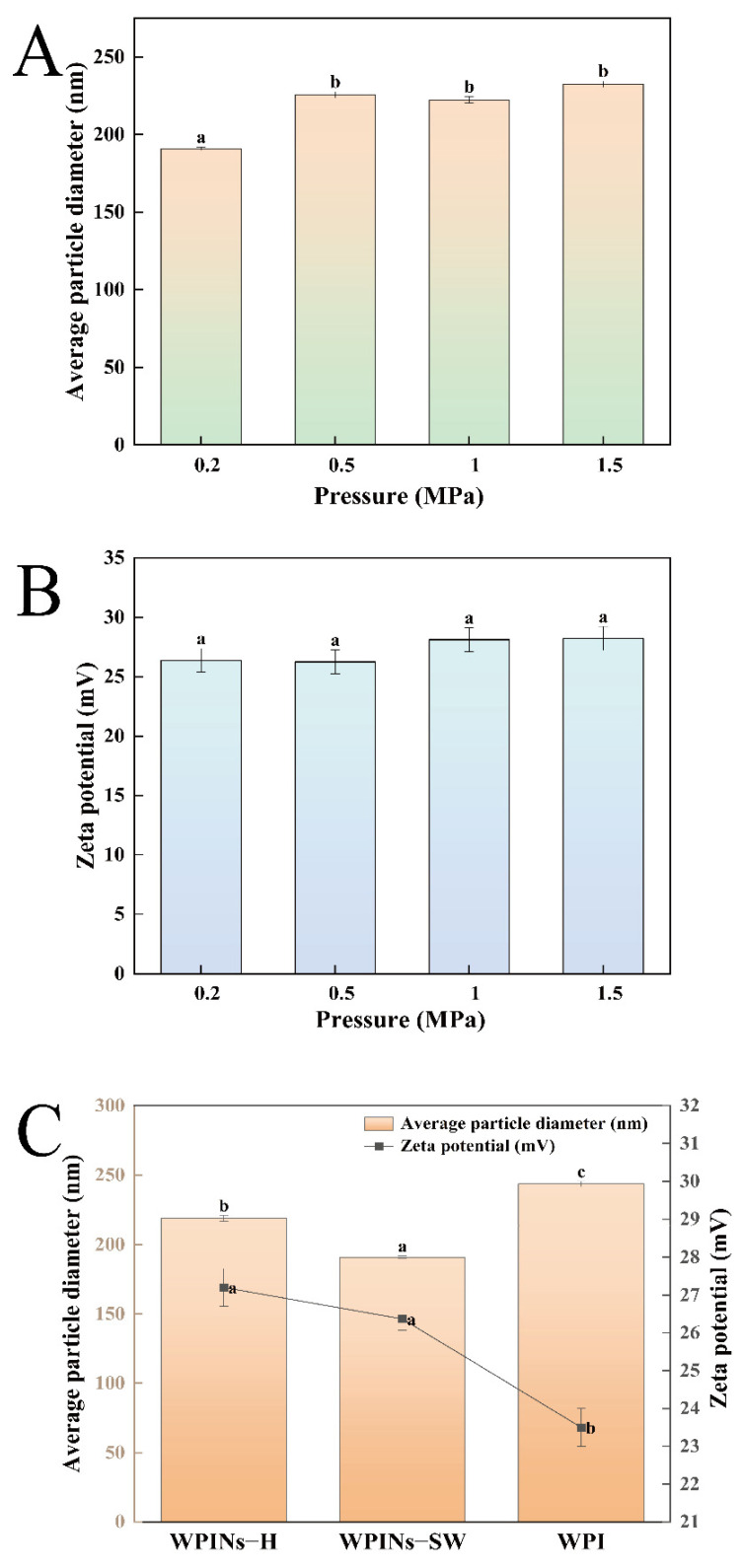
Effects of subcritical water on the structural properties of WPINs. (**A**) The impact of subcritical water pressure on the particle size of WPINs; (**B**) the effect of subcritical water pressure on the zeta potential of WPINs; (**C**) the effect of hydrothermal method and subcritical water method on the particle size and zeta potential of WPINs; (**D**) Congo Red analysis; (**E**) circular dichroism (CD) analysis; (**F**) transmission electron microscopy (TEM) analysis). The different lowercase letters mean that the variance of different samples is significant (*p* < 0.05). Note: WPINs–H represents whey protein isolate nanofibers prepared from hydrothermal method; WPINs–SW represents whey protein isolate nanofibers prepared from subcritical water; WPI represents whey protein isolate.

**Figure 2 foods-11-01625-f002:**
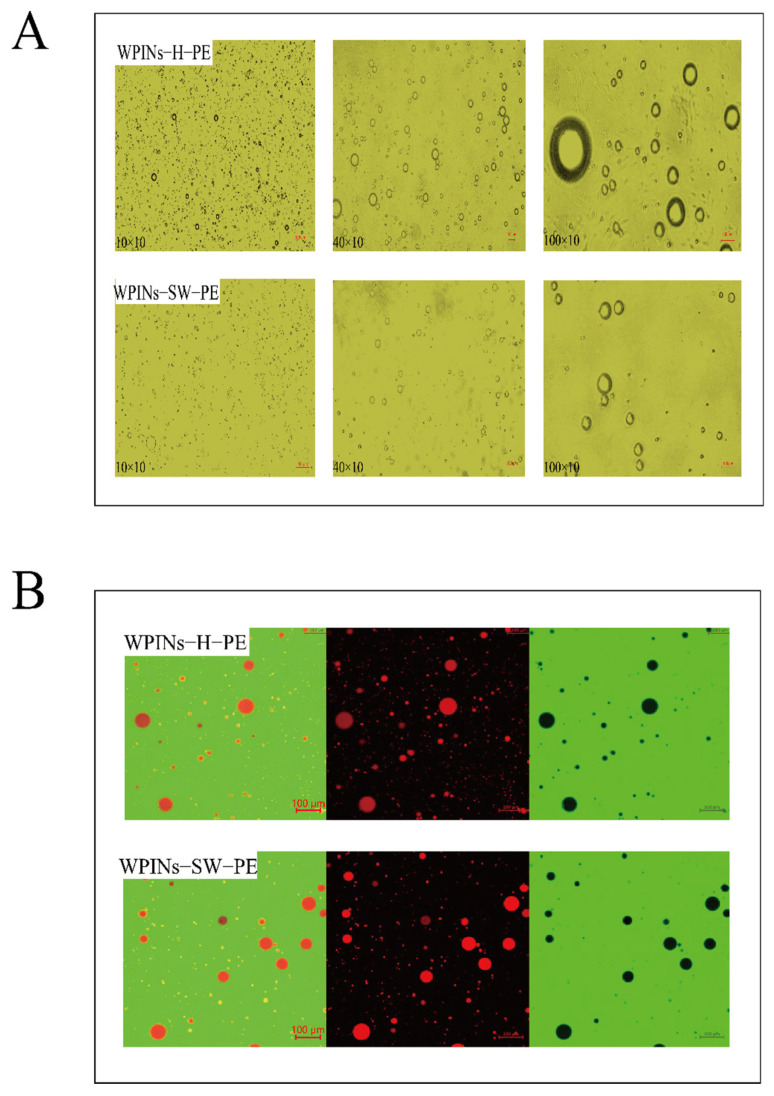
Effects of subcritical water on the properties of high internal phase Pickering emulsion stabilized by WPINs. (**A**) The optical microscope characterization of Pickering emulsion; (**B**) the CLSM imaging of Pickering emulsion; (**C**) the particle size and zeta potential of Pickering emulsion; (**D**) the EAI and ESI of Pickering emulsion; (**E**) the centrifugal stability of Pickering emulsion. The different lowercase letters mean that the variance of different samples is significant (*p <* 0.05). Note: WPINs–H–PE represents whey protein isolate nanofibers prepared from hydrothermal method stabilized Pickering emulsion; WPINs–SW–PE represents whey protein isolate nanofibers prepared from subcritical water stabilized Pickering emulsion.

**Figure 3 foods-11-01625-f003:**
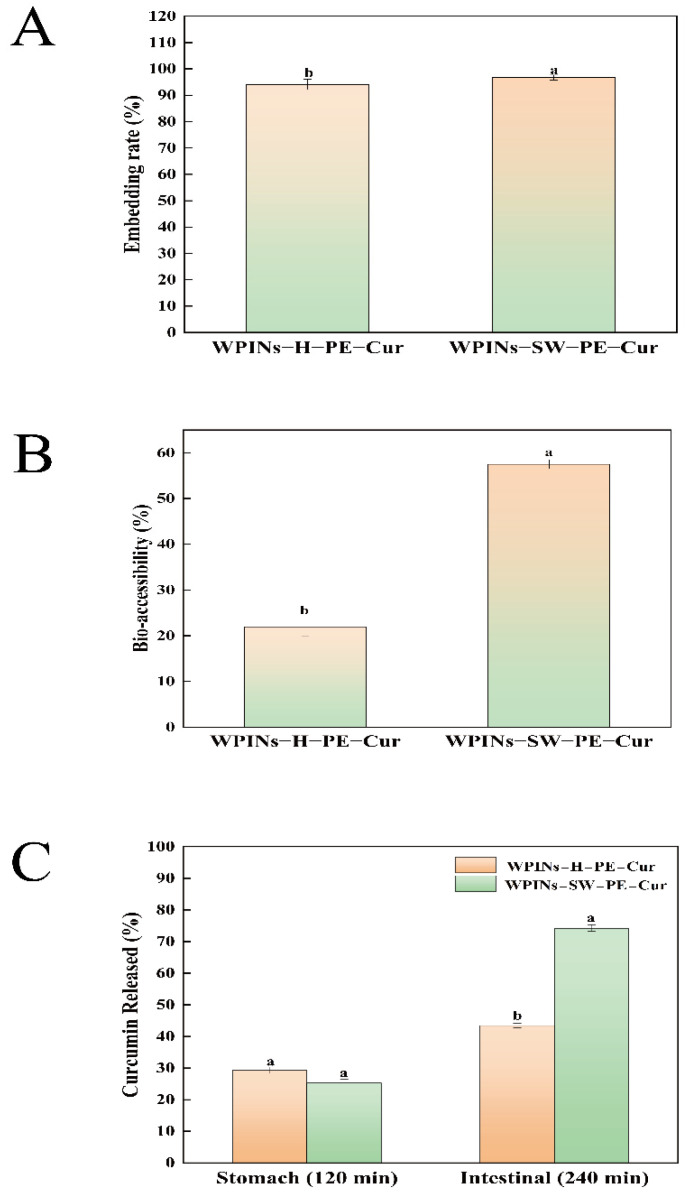
Effects of high internal phase Pickering emulsion on curcumin bioavailability. (**A**) The effect of Pickering emulsion on the embedding rate of curcumin; (**B**) the bio-accessibility of curcumin in Pickering emulsion during gastrointestinal digestion; (**C**) the release rate of curcumin during gastrointestinal digestion. The different lowercase letters mean that the variance of different samples is significant (*p* < 0.05). Note: WPINs–H–PE–Cur represents curcumin-loaded Pickering emulsion prepared from hydrothermal method; WPINs–SW–PE–Cur represents curcumin-loaded Pickering emulsion prepared from subcritical water.

**Figure 4 foods-11-01625-f004:**
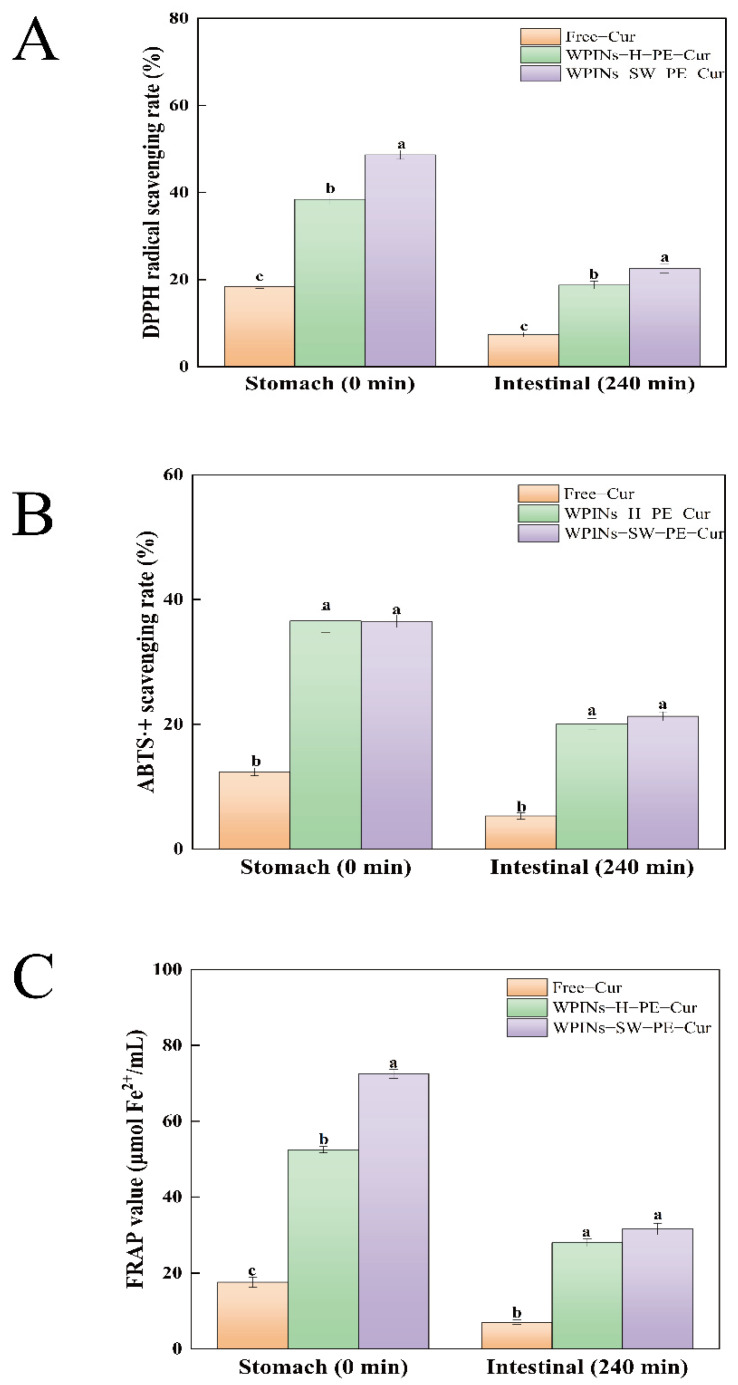
Effects of simulated gastrointestinal digestion on the antioxidant capacity of Pickering emulsion. (**A**) DPPH• scavenging capacity on the Pickering emulsion; (**B**) ABTS·+ scavenging capacity on the Pickering emulsion; (**C**) reducing power on the Pickering emulsion. The different lowercase letters mean that the variance of different samples is significant (*p* < 0.05). Note: WPINs–H–PE–Cur represents curcumin-loaded Pickering emulsion prepared by hydrothermal method; WPINs–SW–PE–Cur represents curcumin-loaded Pickering emulsion prepared by subcritical water.

## Data Availability

Data are contained within the article.

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
