# Peer review of "Whey Protein Isolate Nanofibers Prepared by Subcritical Water Stabilized High Internal Phase Pickering Emulsion to Deliver Curcumin"

_foods, 2022, doi:10.3390/foods11111625_

Round 1
Reviewer 1 Report
This manuscript is interesting. However, the authors give their results without any deep discussions through the manuscript. Therefore, the authors should improve their manuscript with a good point of view. In addition, it is mandatory the revision of the English standard of the paper by a native colleague or a professional.
The detailed comments are listed as follows:
- The keywords should be words that describe the work but you did not manage to include in the title. The aim of keywords is to increase visibility of your article in the databases.
- The authors should mention the current literature in the introduction and more express the novelty of this research.
- A space should insert between number and unit (except °).
- All abbreviations in the figures and tables should be defined in figure captions and table explanations.
Author Response
Comment 1: This manuscript is interesting. However, the authors give their results without any deep discussions through the manuscript. Therefore, the authors should improve their manuscript with a good point of view. In addition, it is mandatory the revision of the English standard of the paper by a native colleague or a professional.
Answer 1: Thank you for your valuable comments. We had re-discussed the results and added new literature. The language of the manuscript had been revised and polished by native English speakers.
Comment 2: The keywords should be words that describe the work but you did not manage to include in the title. The aim of keywords is to increase visibility of your article in the databases.
Answer 2: Thanks, we had modified the keywords (Page: 1, Line: 28-29).
Comment 3: The authors should mention the current literature in the introduction and more express the novelty of this research.
Answer 3: Thanks, we had mentioned the current literature in the introduction to further express the novelty of this manuscript (Page: 2, Line: 47-57, 70-82).
Comment 4: A space should insert between number and unit (except °).
Answer 4: Thanks, we had checked and modified it.
Comment 5: All abbreviations in the figures and tables should be defined in figure captions and table explanations.
Answer 5: Thanks, we had modified it (Page: 9, Line: 320-322; Page: 13, Line: 445-448; Page: 15, Line: 497-500; Page: 17, Line: 549-551).
Thank you very much for your professional comments. Please see the attachment.

Reviewer 2 Report
Article, titled “Whey protein isolate nanofibers prepared by subcritical water 2 stabilized high internal phase Pickering emulsion to deliver 3 curcumin”, talks about using the subcritical water to prepare Pickering emulsion stabilized by whey protein isolate nanofibers. The product was used to encapsulate and prevent curcumin degradationThe text is well written, the topic is interesting, and the large amounts of data are presented in a rather organized manner. The grammar needs some more polishing.
One major thing that I would like to comment on is the length of the abstract (too long); only the most important information should be presented.
Long abstract. The first 2 sentences (“Since the high internal phase Pickering emulsion stabilized ……….. has attracted widespread attention.”) should be deleted
Line 73…..References should be added after “….subcritical water (120°C) heated protein” sentence
Lines 78-79: The sentence; “The aim of this study was to develop subcritical water and a fixed pressure as a new preparation method for the efficient preparation of whey protein isolate nanofibers (WPINs” is confusing, please rewrite it
Line 85 and 86. Please provide more specifications about the whey protein and Corn oil product (lot…etc)
Line 87 “Other reagents”. Which ones?. Please add the reagents used.
2.3.1. Particle size and zeta potential
Both the measurements were performed separately? Why?
Line 104. “vibrates”. Please be more specific
Line 284: 3.1.1. Particle size and zeta potential of whey protein isolate nanofibers
How about the WPI particle size and zeta potential (starting solution)?. The results should be included for comparison. what is the original size of the particles (WPI (starting solution))? And how much decreased
The same for 3.1.2. Congo Red binding spectrum; 3.1.3. Circular dichroism (CD) spectroscopy
Line 335: WPI nanoparticle
Line 347: “…. presented in the figure.” Please added the figure number
Line 366: “ The average particle size and zeta potential of WPINs stabilized high internal phase Pickering emulsion were shown in Fig 2C”…How?
The sentence is confusing, please rewrite it.
Line 441-443. The sentence should be deleted
Figures- the meaning of WPINs-SW and WPINs-H abbreviations should be included in all the figure caption
Author Response
Comment 1: The text is well written, the topic is interesting, and the large amounts of data are presented in a rather organized manner. The grammar needs some more polishing.
Answer 1: Thanks for your kind comments. We had improved the language of the whole manuscript.
Comment 2: One major thing that I would like to comment on is the length of the abstract (too long); only the most important information should be presented. Long abstract. The first 2 sentences (“Since the high internal phase Pickering emulsion stabilized ……….. has attracted widespread attention.”) should be deleted
Answer 2: Thanks, we had deleted inappropriate content and revised the abstract (Page: 1, Line: 16-27).
Comment 3: Line 73…..References should be added after “….subcritical water (120°C) heated protein” sentence
Answer 3: Thanks, we had added the related reference (Page: 2, Line: 80).
Comment 4: Lines 78-79: The sentence; “The aim of this study was to develop subcritical water and a fixed pressure as a new preparation method for the efficient preparation of whey protein isolate nanofibers (WPINs” is confusing, please rewrite it
Answer 4: Thanks, we had revised it (Page: 2, Line: 84-85).
Comment 5: Line 85 and 86. Please provide more specifications about the whey protein and Corn oil product (lot…etc). Line 87 “Other reagents”. Which ones?. Please add the reagents used.
Answer 5: Thanks, we had modified it (Page: 2, Line: 90-98).
Comment 6: 2.3.1. Particle size and zeta potential. Both the measurements were performed separately? Why?
Answer 6: Thanks. There are differences in solution concentrations and vessels required for the determination of particle size and zeta potential. The determination of particle size requires protein particles to be diluted 1000-fold and detected with a four-sided cuvette. The zeta potential was then determined by diluting the protein particles 100-fold and using a potential sample cell.
Comment 7: Line 104. “vibrates”. Please be more specific
Answer 7: Thanks, we had modified it (Page: 3, Line: 117).
Comment 8: Line 284: 3.1.1. Particle size and zeta potential of whey protein isolate nanofibers. How about the WPI particle size and zeta potential (starting solution)?. The results should be included for comparison. what is the original size of the particles (WPI (starting solution))? And how much decreased. The same for 3.1.2. Congo Red binding spectrum; 3.1.3. Circular dichroism (CD) spectroscopy
Answer 8: We had revised it (Page: 7, Line 305-311; Page: 9-10, Line 326-335, 344-348).
Comment 9: Line 335: WPI nanoparticle. Line 347: “…. presented in the figure.” Please added the figure number
Answer 9: We had revised it (Page: 10, Line 355-358, 371).
Comment 10: Line 366: “ The average particle size and zeta potential of WPINs stabilized high internal phase Pickering emulsion were shown in Fig 2C”…How? The sentence is confusing, please rewrite it.
Answer 10: Thanks, we had revised it (Page: 11, Line 400-402).
Comment 11: Line 441-443. The sentence should be deleted
Answer 11: We had deleted it.
Comment 12: Figures- the meaning of WPINs-SW and WPINs-H abbreviations should be included in all the figure caption
Answer 12: Thanks, we had revised it (Page: 9, Line: 320-322; Page: 13, Line: 445-448; Page: 15, Line: 497-500; Page: 17, Line: 549-551).
Thank you very much for your professional comments. Please see the attachment.

Reviewer 3 Report
I reviewed the manuscript entitled, Whey protein isolate nanofibers prepared by subcritical water stabilized high internal phase Pickering emulsion to deliver curcumin. The manuscript has novelty and provided a way to deliver curcumin.
Line 27: were 96.72 ± 1.05% and 94.07 ± 2.35%, respectively??
Line 29: in vitro should be in Italics
Line 41: Curcuma longa should be in Italics
Line 53: G should be in small letter
2.1. Materials and chemicals: what about curcumin? Are authors extract curcumin or purchased from suppliers?
Line 95: certain atmospheric pressure? How much?
Line 95: sentence should not start with number. Please revise as WPI (5%; w/v) solution. Also check throughout the manuscript.
2.3.1. Particle size and zeta potential: add citation
Line 134: 20,000 r/min.. should be revised as 20,000 rpm
Line 147: remove full stop after the reference Li et al. [28]
According to Han et al. [31], According to Liang et al. [32], and According of Gond et al. [30]. Also, check other similar types of citations does not make any sense. Please revise and cite appropriately. For example, X was followed to determine the Y of samples.
Line 210: 30 g of the chyme sample…. Should be revised as chyme sample (30 g)
Lines 217-218: In the light of Zhou et al. [33] The in vitro bio-accessibility of Cur was taken to be the 217 fraction solubilized in the gastrointestinal fluids….. incomplete sentence. Is it in light of Zhou et al. [33], the in vitro……? In vitro should be in Italics
Line 230: UV RANGE?
Line 238: remove in
Include section 2.9. Statistical analysis
Line 335: As shown Fig 1F… should be revised as shown Fig 1F., WPC…….
Results are discussed well with appropriate scientific literature
References
References should be revised according to journal format
Scientific names must be in Italics
Author Response
Comment 1: Line 27: were 96.72 ± 1.05% and 94.07 ± 2.35%, respectively??
Answer 1: Thanks for your nice comment. The encapsulation efficiencies of WPINs-SW-PE (whey protein isolate nanofibers prepared from subcritical water stabilized Pickering emulsion) and WPINs-H-PE (whey protein isolate nanofibers prepared from hydrothermal method stabilized Pickering emulsion) for curcumin were 96.72 ± 1.05% and 94.07 ± 2.35%, respectively, which had been confirmed through three repeated experiments. Although there was no significant difference in the encapsulation efficiency of WPINs-SW-PE and WPINs-H-PE, WPINs-SW-PE had significant advantages in terms of preparation efficiency, bioavailability, intestinal release rate, and antioxidant activities. The specific results were as follows:(1) The encapsulation efficiencies of protein nanofiber stabilized Pickering emulsions obtained by conventional hydrothermal and subcritical water treating whey protein isolate for 10 h and 5 min had little difference, which indicated that the preparation efficiency of subcritical water was significantly higher than that of hydrothermal method. (2) Compared with WPINs-H-PE, the bio-accessibility, intestinal release rate, and antioxidant activity of curcumin-loaded WPINs-SW-PE were increased by 35.58%, 30.8%, and 10.22%, respectively. In conclusion, WPINs-SW-PE had better curcumin delivery advantages and application prospects in comparison with WPINs-H-PE.
Comment 2: Line 29: in vitro should be in Italics. Line 41: Curcuma longa should be in Italics. Line 53: G should be in small letter
Answer 2: Thanks, we had revised it (Page: 1, Line: 32).
Comment 3: 2.1. Materials and chemicals: what about curcumin? Are authors extract curcumin or purchased from suppliers?
Answer 3: Curcumin was purchased from a supplier and we had modified it in the manuscript (Page: 2, Line 90).
Comment 4: Line 95: certain atmospheric pressure? How much?
Answer 4: The atmospheric pressure is 0.2 MPa, 0.5 MPa, 1.0 MPa, and 1.5 MPa. We had revised it in the manuscript (Page: 3, Line 107-108).
Comment 5: Line 95: sentence should not start with number. Please revise as WPI (5%; w/v) solution. Also check throughout the manuscript.
Answer 5: Thanks, we had checked the whole manuscript and modified it (Page: 3, Line: 110).
Comment 6: 2.3.1. Particle size and zeta potential: add citation
Answer 6: Thanks, we had added it (Page: 3, Line: 114-115).
Comment 7: Line 134: 20,000 r/min.. should be revised as 20,000 rpm
Answer 7: Thanks, we had revised it (Page: 4, Line: 147).
Comment 8: Line 147: remove full stop after the reference Li et al. [28]
According to Han et al. [31], According to Liang et al. [32], and According of Gond et al. [30]. Also, check other similar types of citations does not make any sense. Please revise and cite appropriately. For example, X was followed to determine the Y of samples.
Answer 8: Thanks, we had revised it (Page: 4, Line: 160, 173, 182; Page: 5, Line: 195, 223, 226).
Comment 9: Line 210: 30 g of the chyme sample…. Should be revised as chyme sample (30 g)
Answer 9: Thanks, we had revised it (Page: 5, Line: 218).
Comment 10: Lines 217-218: In the light of Zhou et al. [33] The in vitro bio-accessibility of Cur was taken to be the 217 fraction solubilized in the gastrointestinal fluids….. incomplete sentence. Is it in light of Zhou et al. [33], the in vitro……? In vitro should be in Italics
Answer 10: Thanks, we had revised it (Page: 5, Line: 225-226).
Comment 11: Line 230: UV RANGE?
Answer 11: The wavelength of UV was set as 425 nm, we had added it in the manuscript (Page: 6, Line: 239).
Comment 12: Line 238: remove in
Answer 12: Thanks, we had deleted it.
Comment 13: Include section 2.9. Statistical analysis
Answer 13: Thanks, we had added it (Page: 7, Line: 278-282).
Comment 14: Line 335: As shown Fig 1F… should be revised as shown Fig 1F., WPC…….
Answer 14: Thanks, we had added it (Page: 9, Line: 320-322; Page: 13, Line: 445-448; Page: 15, Line: 497-500; Page: 17, Line: 549-551).
Comment 15: References should be revised according to journal format. Scientific names must be in Italics.
Answer 15: We had revised it.
Thank you very much for your professional comments.
Round 2
Reviewer 3 Report
Based on the author responses, the present version of the manuscript can be accepted for publication.